



# Earth System Model Parameter Adjustment Using a Green's Functions Approach

Ehud Strobach[1,2], Andrea Molod[2], Donifan Barahona[2], Atanas Trayanov[2,3], Dimitris Menemenlis[5], and Gael Forget[4]

[1]Agricultural Research Organization, Rishon Lezion, Israel
[2]Goddard Space Flight Center, Greenbelt, MD, US
[3]Science Systems and Applications Inc., Greenbelt, MD, US
[4]Massachusetts Institute of Technology, Cambridge, MA, US
[5]Jet Propulsion Laboratory,California Institute of Technology, Pasadena, CA, US

**Correspondence:** Ehud Strobach (udist@volcani.agri.gov.il)

**Abstract.** We demonstrate the practicality and effectiveness of using a Green's functions estimation approach for adjusting uncertain parameters in an Earth System Model (ESM). This estimation approach had previously been applied to an intermediate-complexity climate model and to individual ESM components, e.g., ocean, sea-ice, or carbon-cycle components. Here, the Green's functions approach is applied to a state-of-the-art ESM that comprises a global atmosphere-land configu-

ration of the Goddard Earth Observing System (GEOS) coupled to an ocean and sea-ice configuration of the Massachusetts Institute of Technology general circulation model (MITgcm). Horizontal grid spacing is approximately 110 km for GEOS and 37–110 km for MITgcm. In addition to the reference GEOS-MITgcm simulation, we carry out a series of model sensitivity experiments, in which 20 uncertain parameters are perturbed. These "control" parameters can be used to adjust sea-ice, microphysics, turbulence, radiation, and surface schemes in the coupled simulation. We define eight observational targets: sea-ice

fraction, net surface shortwave radiation, downward longwave radiation, near-surface temperature, sea surface temperature, sea surface salinity, and ocean temperature and salinity at 300 m. We applied the Green's functions approach to optimize the values of the 20 control parameters so as to minimize a weighted least-squares distance between the model and the eight observational targets. The new experiment with the optimized parameters resulted in a total cost reduction of $9\%$ relative to a simulation that had already been adjusted using other methods. The optimized experiment attained a balanced cost reduction over most of

the observational targets. We also report on results from a set of sensitivity experiments that are not used in the final optimized simulation but helped explore options and guided the optimization process. These experiments include an assessment of sensitivity to the number of control parameters and to the selection of observational targets and weights in the cost function. Based on these sensitivity experiments, we selected a specific definition for the cost function. The sensitivity experiments also revealed a decreasing overall cost as the number of control variables was increased. In summary, we recommend using

the Green's functions estimation approach as an additional fine-tuning step in the model development process. The method is not a replacement for modelers' experience in choosing and adjusting sensitive model parameters. Instead, it is an additional practical and effective tool for carrying out final adjustments of uncertain ESM parameters.



# 1   Introduction

Earth System Models (ESMs) include various parameters that govern the representation of unresolved, unrepresented, or
under-observed processes in the models. The most sensitive parameters are typically adjusted during the last step of model
development relative to observational targets. Currently, there is no agreed-upon methodology to adjust model parameters. As
an illustration of the range of approaches and observational targets, Schmidt et al. (2017) describe the vast range of tuning
strategies used for ESM development in U.S. modeling centers. Parameter adjustment aims to improve the representation of
various processes or key state fields in the model relative to a pre-determined set of observational targets (Mauritsen et al.,
2012; Hourdin et al., 2017). Typically, the tuning is done in a sequential manner in which one or more parameters are per-
turbed to new values based on the diagnosis of the causes of systematic error, with the aim of improving the model's overall
behavior (e.g., Watanabe et al., 2010; Yukimoto et al., 2012; Mauritsen et al., 2019; Voldoire et al., 2019; Golaz et al., 2019;
Sellar et al., 2019; Danabasoglu et al., 2020). The observational targets can include, for example, radiation balance at the top
of the atmosphere, surface temperature, sea ice variability, tropospheric wind, or even a derived target such as a "cloud object"
(Posselt et al., 2015). Sequential ESM tuning involves running many model sensitivity experiments, where one or more model
parameters have been perturbed and comparing the results with one or more observational targets. This tuning exercise is highly
dependent on the experience and expertise of model developers, and on the application for which the model will be used.

In addition to sequential tuning methods with a subjective choice of the parameter values for the sensitivity suite, there
are automatic parameter calibration methods that have been demonstrated, in proof of concept studies, to more systematically
choose or "fine-tune" the final set of optimized parameters. These methods include the use of Latin Hypercube sampling tech-
niques (Posselt et al., 2015), the downhill simplex method (Zhang et al., 2015), and the multiple very fast simulated annealing
(Zou et al., 2014). As is the case for all of the subjective and automatic tuning methods, the choice of parameter space to
explore is based on the diagnosis of errors and the modelers' intuition. Both the subjective and automatic parameter estimation
methods require a large suite of experiments, which are computationally expensive and therefore have been performed with
very low resolution, atmosphere-only, or even single-column experiments. The direct applicability of these methods for tuning
an ESM may consequently be limited.

One major issue related to sequential tuning, whether using subjective or automatic methods, is the interdependence of
each parameter's effect on the resulting ESM simulation and the relationship to observational targets. For example, both sea
ice albedo and precipitation efficiency could affect the globally averaged radiation balance at the top of the atmosphere. An
experiment with both perturbed would not reveal which had the greater impact on the solution, and experiments with only one
at a time perturbed would not behave as an experiment with both perturbed. The sequential methodology, therefore, for this
example, may result in a sub-optimal combination of these parameters.

Another issue related to sequential tuning is the computational cost of running a suite of modern ESMs. A typical sequential
tuning exercise includes a series of sensitivity experiments with one or more perturbed parameters, a comparison against a set
of observations, and another set of sensitivity experiments targeting either the same or another set of observations based on
the results. There are two main drawbacks of the sequential nature of this exercise. The first is that this method necessitates an



analysis step between each sensitivity sweep, a time-consuming process. The second drawback is the large number of needed simulations, since a new set of experiments is required for each set of observational targets, and may also include several experiments perturbing the same parameter or set of parameters. Given unrestricted computer resources, one could explore

model parameter space exhaustively and choose the simulation that best fits all available observations. In practice, this type of exhaustive parameter exploration is not feasible, and we need tractable methods that combine objective methodologies with the model developer's experience to arrive at the optimal choice of model parameters.

One alternative to the sequential method to tune models is the adjoint methodology. When using this methodology, one can adjust many parameters simultaneously by running one experiment. A successful example is the ECCO state estimate that

used the adjoint methodology to adjust MITgcm ocean model parameters, initial conditions, and boundary conditions (Forget et al., 2015). Although this methodology provides robust results, developing a model's adjoint is onerous, even when using a software tool, and using it is computationally expensive.

In this study, we use a Green's functions methodology (Menemenlis et al., 2005) for the first time to tune a fully coupled model - the GEOS-MITgcm ESM. The Green's functions methodology addresses the major issues with sequential tuning

identified above, and has some features of the adjoint methodology. The Green's functions methodology can adjust several parameters simultaneously while accounting for the potential interdependence of their effect on the simulation. The interdependence is accounted for in the constraints on the error covariance for each parameter. In addition, some of the computational expense of the sequential tuning method is alleviated, since only one set of sensitivity experiments is required, so there is little or no expense relative to an extensive range of observational targets or combinations of parameters. The no-iteration character-

istic is also an advantage relative to the Gauss–Newton line search algorithm (Tett et al., 2013, 2017; Roach et al., 2018) which uses similar aproach to minimized a set of parameters simultaneously.

The Green's functions methodology does assume that the effect of the perturbed parameter is approximately linear. It can also incorporate as many observational targets (with negligible computational cost) as desired as long as data constraints are available. The results described here include a proof of concept for the use of the methodology in an ESM and the examination

of various 'flavors' of the optimization process. The mathematical formalism and the Green's functions method are presented in the next section. This presentation is followed by the description of the model, the experiments, and the used observational targets. Section 4 presents our chosen configuration results, followed by a Section outlining the motivation for the configuration choices. The study is summarized in Section 6.

## 2   Earth System Model Description

The model used for this study is the GEOS-MITgcm coupled model. We briefly describe here the particular configurations of these two models as they are used in our study. The GEOS AGCM's dynamical core and suite of physical parameterizations, along with the land model and aerosol model are described in Molod et al. (2020). The turbulent surface layer parameterization, relevant for this study, is a modified version of the parameterization documented in Helfand and Schubert (1995). The wind stress and surface roughness model was modified by the updates of Garfinkel et al. (2011) for a mid-range of wind speeds and





further modified by the updates of Molod et al. (2013) for high winds. The GEOS AGCM's cubed-sphere grid was configured
to run with a nominal horizontal grid spacing of 100 km. The vertical grid is a hybrid sigma-pressure grid with 72 levels.
MITgcm has a finite-volume dynamical core (Marshall et al., 1997). It has a nonlinear free-surface and real freshwater flux
(Adcroft and Campin, 2004) and a nonlocal K-profile parameterization scheme for mixing (Large et al., 1994). The MITgcm
horizontal grid is the so-called "Lat-Lon-Cap" (Forget et al., 2015), and was configured to run with a nominal grid spacing
of 100 km. The vertical grid is the $z^*$ height coordinates (Adcroft and Campin, 2004), and it has 50 vertical levels. The
GEOS-MITgcm atmosphere-ocean interface includes a skin layer (Price et al., 1978), configured for the simulations described
here to impose a one-day time scale on the interaction, and the communication between the ocean and atmosphere is updated
at every atmospheric model time step (450 seconds). The sea-ice dynamics model is provided by MITgcm and the sea ice
thermodynamics model is from CICE 4.1 (Hunke and Lipscomb, 2010).

## 3  Mathematical Formalism and Methodology

Algebraically, the ESM described in previous section can be expressed as a set of rules for time stepping a state vector,

$$\boldsymbol{x}(t_{i+1}) = M_i(\boldsymbol{x}(t_i)), \tag{1}$$

where $\boldsymbol{x}(t_i)$ is a discretized representation of the Earth System at time $t_i$ and function $M_i$ represents ESM time-stepping rules
for advancing the state vector from time $t_i$ to $t_{i+1}$. The discretized evolution of the true Earth System, $\boldsymbol{x}^t(t_i)$, is assumed to
differ from that of the numerical model by a vector of stochastic perturbations $\boldsymbol{\eta}$ so that

$$\boldsymbol{x}^t(t_{i+1}) = M_i(\boldsymbol{x}^t(t_i), \boldsymbol{\eta}), \tag{2}$$

where $\boldsymbol{\eta}$ is assumed to have zero mean and covariance matrix $\mathbf{Q}$. Consider a set of observations represented by vector $\boldsymbol{y}^o$,
which are related to the discretized, time-evolving Earth System state vector by measurement function $H$, that is,

$$\boldsymbol{y}^o = H(\boldsymbol{x}^t(t_0), \boldsymbol{x}^t(t_1), \ldots, \boldsymbol{x}^t(t_N)) + \boldsymbol{\varepsilon}, \tag{3}$$

where $\boldsymbol{\varepsilon}$ represents a vector noise process assumed to have zero mean and covariance matrix $\mathbf{R}$. In addition to measurement
errors, vector $\boldsymbol{\varepsilon}$ includes all model and estimation "representation" errors, i.e., model-data residuals that cannot be represented
by $M_i$ and $\boldsymbol{\eta}$. The optimization problem aims to minimize an objective cost function

$$J = \boldsymbol{\eta}^T \mathbf{Q}^{-1} \boldsymbol{\eta} + \boldsymbol{\varepsilon}^T \mathbf{R}^{-1} \boldsymbol{\varepsilon}, \tag{4}$$

which penalizes the least-squares distance relative to model priors and observations. For the Green's Functions approach, we
adjust a small, carefully-chosen set of parameters $\boldsymbol{\eta}$. To do this, we combine Eqs. (2) and (3) to obtain:

$$\boldsymbol{y^o} = G(\boldsymbol{\eta}) + \boldsymbol{\varepsilon}, \tag{5}$$





where $G$ represents the convolution of observation operator $H$ with ESM time-stepping rules $M_i$. We assume that (5) can be linearized about a reference ESM simulation so that:

$$\boldsymbol{y}^d = \boldsymbol{y}^o - G(\mathbf{0}) = \mathbf{G}\boldsymbol{\eta} + \boldsymbol{\varepsilon}, \tag{6}$$

where $\mathbf{0}$ is a null vector, $G(\mathbf{0})$ represents the reference ESM simulation sampled by measurement function $H$, vector $\boldsymbol{y}^d$ is the model-data difference between observations and the reference simulation, and kernel matrix $\mathbf{G}$ can be derived by conducting an ESM perturbation experiment for each element of vector $\boldsymbol{\eta}$. Specifically, the $j$-th column of matrix $\mathbf{G}$ is

$$\boldsymbol{g}_j = \frac{G(\boldsymbol{e}_j) - G(\mathbf{0})}{e_j}, \tag{7}$$

where $\boldsymbol{e}_j$ is a perturbation vector filled with zeros except for element $j$, which is perturbed by scalar value $e_j$. The minimization

of (4) given (5) is a discrete linear inverse problem with solution:

$$\boldsymbol{\eta}^a = \mathbf{P}\mathbf{G}^T\mathbf{R}^{-1}\boldsymbol{y}^d, \tag{8}$$

where

$$\mathbf{P} = (\mathbf{Q}^{-1} + \mathbf{G}^T\mathbf{R}^{-1}\mathbf{G})^{-1} \tag{9}$$

is the uncertainty covariance matrix for the optimized estimate $\boldsymbol{\eta}^a$. In practice, the Green's functions estimation method can be

divided into six steps:

1. Run a reference ESM simulation using default values for model-error parameters, that is, $\boldsymbol{\eta} = \mathbf{0}$. The output from this experiment, sampled by observation operator $H$, provides $G(\mathbf{0})$ in (5) and (6).

2. Choose a set of observational targets, that is, define operator $H$ and vector $\boldsymbol{y}^o$ in (3).

3. Estimate error covariance matrices $\mathbf{Q}$ and $\mathbf{R}$, the model and observation/representation errors, respectively, that are

needed to define cost function $J$ in (4).

4. Run a set of model sensitivity experiments, where each element of vector $\boldsymbol{\eta}$ is perturbed, one at a time, by scalar value $e_j$. These sensitivity experiments are used to construct kernel matrix $\mathbf{G}$ as per recipe given by Eq. (7). This step holds the biggest computational cost, far exceeding other steps.

5. Calculate a set of optimized model error parameters $\boldsymbol{\eta}^a$ using Eq. (8). Vector $\boldsymbol{\eta}^a$ can be used to compute an optimized

linear combination and a *projected cost* under the assumption of linearity.

6. Run a new simulation using optimized parameters $\boldsymbol{\eta}^a$. This *optimized simulation* can be compared with the optimized linear combination of sensitivity experiments in order to evaluate the linearization assumption.

Note that steps the choice of the observational targets and the definition of the covariance matrices (steps 2 and 3) can be later revised without the need to rerun a new set of experiments (step 4).



## 4  Proof of Concept Optimization

In this section, we illustrate the application of the six steps listed above.

### 4.1  Reference Experiment

The 10-year long reference experiment was configured with reference values of a set of parameters that we aimed to optimize. The model was configured to run in "perpetual year" mode, meaning that the external boundary conditions (solar insolation, greenhouse gas amount and aerosol emissions ) are kept to those of 2000. The "perpetual year" mode was chosen to optimize the model's equilibrium state relative to our climatology observational targets.

### 4.2  Observational targets

**Table 1.** List of observational targets and covariance value.

| Variable | Dataset | Years | Covariance (R) |
|---|---|---|---|
| Seaice fraction | MERRA-2[1] | 1996-2004 | 0.14 |
| Net surface shortwave radiation | SRB[2] | 1996-2004 | 2250 [Wm-2]$^2$ |
| Downward longwave radiation | SRB[2] | 1997-2004 | 506 [Wm-2]$^2$ |
| Near surface temperature | HadCRUT4[3] | 1997-2004 | 35.4 [°C]$^2$ |
| Sea surface temperature | ECCO[4] | 1996-2004 | 6.67 [°C]$^2$ |
| Sea surface salinity | ECCO[4] | 1996-2004 | 2.06 [psu]$^2$ |
| THETA at 300m | ECCO[4] | 1996-2004 | 2.62 [°C]$^2$ |
| Salt at 300m | ECCO[4] | 1996-2004 | 0.05 [psu]$^2$ |

[1]GMAO (2016) [2]Stackhouse Jr et al. (2011) [3]Osborn and Jones (2014) [4]Forget et al. (2015)

The motivation of this study was to set up a simple, systematic, reproducible, extensible, and efficient framework for improving the climatology of the new coupled model now and as it undergoes development in the future. To start, we chose eight key observational targets (Table 1), each one chosen for reasons outlined below. Ice fraction is directly relevant to all sea ice processes, and is important for an accurate representation of climate variability in the polar regions and climate feedbacks on the global scale. Accurate modeling of shortwave radiation at the surface indicates that the model can simulate the atmospheric and oceanic circulations' driving force. Shortwave radiation and downward longwave radiation indicate the realistic representation of cloud processes in the atmosphere. Near-surface temperature was added aiming to balance out ocean-only observational targets. Sea Surface Temperature (SST) is a fundamental climate variable that drives the atmosphere. Sea Surface Salinity (SSS) controls ocean circulation but also a tracer of the global water cycle. Their equivalents at 300m below sea level additionally constrain large scale ocean transports. The observational targets were calculated separately for each season - DJF, MAM, JJA, SON, and each seasonal mean was compared with corresponding model output.





### 4.3 Definition of cost function

In this study, $\mathbf{Q}^{-1}$ is set to zero, that is, the cost function does not explicitly include *a priori* knowledge about the range of possible parameter values. Some of the parameters, however, have intrinsic physical bounds for the range of values. For these parameters, the resulting optimized parameters were checked and all were found to lie within the range of physically-acceptable values. The error covariance matrix $\mathbf{R}$ is chosen to be diagonal. The variance was calculated separately for each observational target and defined to be the global, area-weighted, mean-squared difference between the climatology of the

reference experiment and observations, that is:

$$R_v = \sum_{i,j,s} w_{i,j} \cdot \left( \hat{y}^o_{v,i,j,s} - \hat{x}^r_{v,i,j,s} \right)^2, \tag{10}$$

where the indices $v$, $i$, $j$, and $s$ represent observation variable, longitude, latitude, and season (winter, spring, summer, and fall), the hat sign ˆ represents climatological seasonal-mean values, and $w_{i,j}$ is an area weight with $\sum_{i,j} w_{i,j} = 1/4$, giving each season a $1/4$ of the weight. The prior error variance of each variable is, therefore, a constant value. A consequence of

our choice of the covariance matrix and the cost function is that the cost of the reference experiment is equal to one for each variable. For eight observational targets, the total cost of the reference experiment is eight. This definition gives equal weight to each of the eight observational targets and was found to provide a balanced solution in terms of each observational target's overall influence. The cost of the optimized experiment is expected to be smaller than one for each variable.

### 4.4 Sensitivity Experiments

In addition to the reference experiment, 20 10-year long sensitivity experiments were performed using the exact model configuration as the reference experiment but perturbing one of the uncertain parameters each time. The reference and perturbed parameter values, and a short description of the parameters is listed in Table 2. We used the 20 sensitivity experiments to estimate the model's sensitivity to each of the perturbed parameters based on the Green's functions formalism. The perturbed parameter values where chosen based on expert advise. For some of the parameters the value was chosen based on physically

realistic values from past experience. In some of the cases, the perturbed value was chosen arbitrarily due to a large parameter uncertainty.

### 4.5 Optimized Parameters

The optimized parameters are given in the last column of Table 2. It can be seen that only half of the of optimized parameter values fall between the reference and the perturbed value, indicating a large unpredictability in model response to the change

in those parameters. Nevertheless, the difference of the optimized parameter and the reference value ($\boldsymbol{\eta}^a$) did not exceed the difference of the perturbed parameteer value and the reference value ($\boldsymbol{\eta}$) except for one parameter - TS_AUTO_ICE. This may indicate that our choice of the perturbed values was within a realistic range. Particularly, when considering that $\mathbf{Q}^{-1}$ was set to zero, suggesting that the optimized parameter's range were not constrained by the cost function.





**Table 2.** List of perturbed parameters

| # | Parameter name | Description | Reference | Perturbed | Optimized |
|---|---|---|---|---|---|
| 1 | AUTscale | Liquid autoconversion rate scaling | 0.20 | 0.30 | 0.27 |
| 2 | LTS_LOW | For LTS higher than LTS_LOW, and in marine regions, the cloud | 19.0 | 20.0 | 19.2 |
| 3 | MIN_EXP | fraction within the PBL is scaled using a power law, where the power | 0.60 | 0.80 | 0.73 |
| 4 | MAX_EXP | exponent varies between MIN_EXP and MAX_EXP. | 1.50 | 1.70 | 1.32 |
| 5 | CICE_AH_MAX | Used in calculating the ice albedo threshold | 0.30 | 0.20 | 0.36 |
| 6 | ALBICEV | Visible band ice albedo | 0.73 | 0.82 | 0.79 |
| 7 | ALBICEI | Near-IR band ice albedo | 0.33 | 0.40 | 0.37 |
| 8 | TURN_RHCRIT | The level at which the PDF of total moisture is assumed to change from its "free atmosphere" behavior to its "inside PBL" behavior. | 884 | 750 | 870 |
| 9 | CQfactor | A factor for the surface moisture exchange coefficient | 1.0 | 1.5 | 1.2 |
| 10 | Charnok1 | Governs the relationship between the ocean wind stress and the wind speed. This parameter governs the behavior at high wind speed. | 2.92E-3 | 2.19E-3 | 3.65E-3 |
| 11 | Charnok2 | Governs the relationship between the ocean wind stress and the wind speed. This parameter governs the behavior at low wind speed. | -1.10E-8 | -2.00E-8 | -1.70E-8 |
| 12 | VSLfactor | Scales the impact of the laminar viscous sublayer over the ocean. | 1.00 | 1.20 | 0.96 |
| 13 | MF2CFfactor | Mass flux to cloud factor - governs the relationship between detrained cloud mass flux and cloud fraction. | 1.00 | 0.80 | 1.14 |
| 14 | MIN_RHCRIT | Determines the width of the assumed PDF of total water in the free troposphere. | 0.90 | 0.85 | 0.91 |
| 15 | TS_AUTO_ICE | Auto-conversion time scale for ice in multiples of model time step | 4.0 | 3.0 | 4.5 |
| 16 | BC_INFAC | The fraction of black carbon number concentration that is allowed to act as ice nucleating particles. | 1.00 | 0.50 | 0.35 |
| 17 | DUST_INFAC | The fraction of dust number concentration that is allowed to act as ice nucleating particles. | 1.00 | 0.50 | 0.79 |
| 18 | DCS | Critical diameter for ice auto-conversion. | 3.5E-4 | 3.0E-4 | 3.8E-4 |
| 19 | UISCALE | Scaling for the terminal velocity of ice | 1.00 | 0.90 | 0.98 |
| 20 | KHRADFAC | Scale factor for negatively buoyant turbulence related to cloud-top cooling | 0.85 | 0.50 | 0.95 |

## 4.6 Assessing the Performance of the Green's Functions Methodology

Once the Green's functions methodology provides the optimized parameters, a projected cost can be derived directly from the sensitivity experiments using the optimized parameters. This derivation can be done before actually running the optimized simulation with the optimized parameters. The projected cost provides a first indication of the expected cost reduction assuming





linearity. After a new optimized experiment is performed, the projected cost can be compared with the optimized experiment's cost to assess the correctness of the assumption of linearity.

Figure 1 (and Table 3) shows that the total cost reduction of the optimized experiment (shown on the right) relative to the reference experiment (shown on the left) is 9% (similar to the cost reduction found in Zhang et al. (2015)). Figure 1 (and Table 3) also shows the projected cost, which is the cost one would get if the model response to parameter perturbations were entirely linear. Most of the cost reduction seen in the projected (second from the right) experiment is realized in the optimized experiment. The total cost of each of the 20 sensitivity experiments is larger than the cost of the simulation with the optimized

parameters. Figure 2 shows the normalized cost for each of the eight observational targets in the same order as in Figure 1. Figure 2 shows that the cost is the lowest in the optimized experiment (the rightmost bar in each cluster) for five out of the eight observational targets. For one of the observational targets (sea-ice fraction) the cost of the optimized experiment is smaller than the reference but several sensitivity experiments have lower cost. For two observational targets, the optimized experiment cost is larger than the cost of the reference experiment. This cost increase happens for surface temperature over land and downward

longwave radiation. The longwave radiation had a cost increase of 2%. The more considerable increased cost of 16% percent in the surface temperature is related to our choice of the observational targets. Five out of the eight observational targets are ocean-only variables, only one is land-only, and two cover both land and ocean. This asymmetry between land and ocean constraints seems to have pulled the parameters to values that benefit the ocean more than the land. This proof of concept illustrated in this study, although clearly demonstrates the overall cost reduction with this method, also illustrates the need to

choose the desired observational targets carefully.

**Table 3.** Explained variance in percentage. Negative values represent reduction of explained variance. TS - Near surface temperature; FR - Seaice fraction; LW - Downward longwave radiation; SW - Net surface shortwave radiation; SST - Sea Surface Temperature; SSS - Sea surface salinity; T (300) - THETA at 300m; S (300) -Salt at 300m.

|           | TS    | FR   | LW   | SW   | SST  | SSS  | T (300m) | S (300m) | Total |
|-----------|-------|------|------|------|------|------|----------|----------|-------|
| Projected | 7.3   | 12.9 | 11.3 | 14.4 | 36.3 | 10.0 | 21.4     | 11.6     | 15.6  |
| Optimized | -16.0 | 8.2  | -2.0 | 20.0 | 25.6 | 14.5 | 11.3     | 10.6     | 9.0   |

The cost reduction was also found to be rather homogeneous across many regions around the globe. Figure 3 shows that, for net shortwave radiation, the total cost reduction was 20% relative to the reference experiment. Most of the oceans exhibit a large reduction in the cost, particularly over the Antarctic Circumpolar Current (ACC) and in the stratocumulus regions in the eastern Pacific's subtropics. In comparison with these cost reductions over the oceans, the land still exhibits marginal increases

in cost over wide regions of North America and south-east Asia.

The actual cost reduction of the various variables seen in the optimized experiment is also generally consistent with the projected cost. Figure 4 shows the projected cost reduction for the net shortwave radiation, which is in broad agreement with the actual cost reduction of the optimized experiment seen in figure 3c. In this case, the cost of the optimized experiment was even smaller than the projected cost. Overall agreement between the projected cost and optimized experiment cost suggests





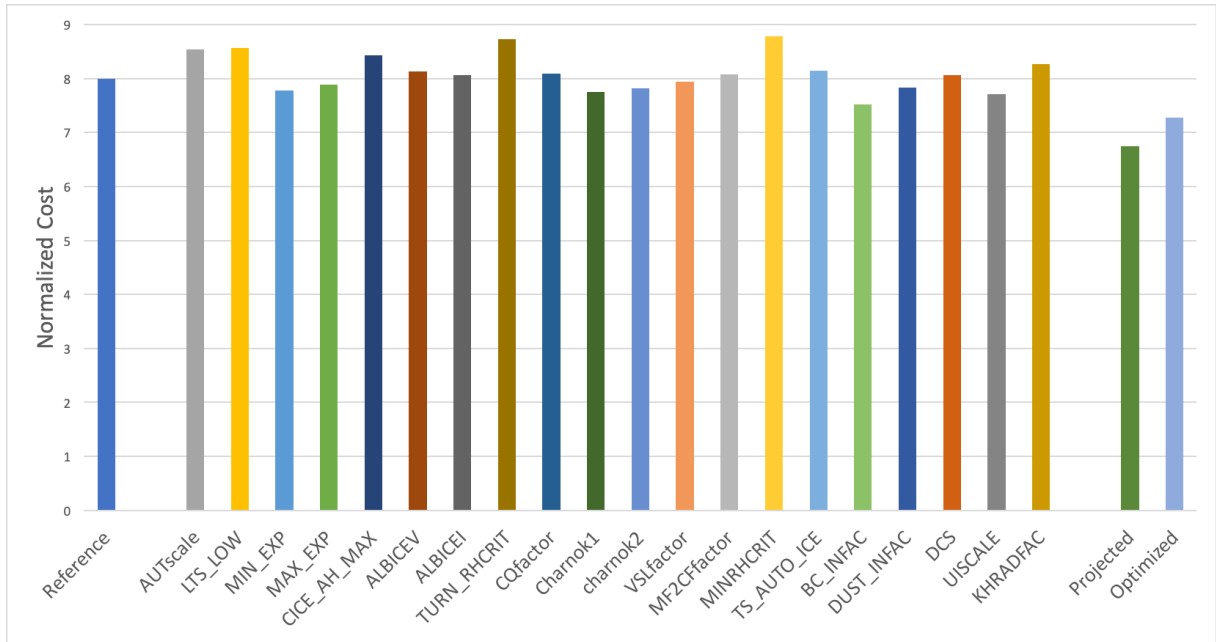

**Figure 1.** The total cost of the reference experiment, the optimized experiment and the 20 perturbed experiments. Projected cost is also indicated.

that the linearity assumption that underlies the Green's functions methodology holds sufficiently well for the approach to be useful.

     The surface temperature, which exhibits an overall 16% increase in cost (Figure 5), also shows some large regions of cost reduction over North America and Greenland. However, the cost in north Asia and Europe is increased. In terms of the cost's seasonality, it was found that DJF and MAM had the highest cost increase while JJA months had the smallest overall cost

reductions. These results suggest that the misrepresentation of land-ice processes in the new and the old configurations is responsible for the high cost. Surface temperature is the variable that had the least projected cost reduction. So, it was expected to exhibit the worst performance in terms of the optimized experiment cost. Downward longwave radiation also exhibits cost increase, but here the increase is 2%. In terms of the other variables, the cost of sea-ice fraction was reduced by 8.2%, and the four ocean-only variables showed cost reduction between 10% to 25%, indicating an improved representation of the ocean

circulation.

## 5   Discussion: Motivation for Choices in Methodology

The parameter estimation experiment presented in the previous section describes our current best practice, which was guided by a series of sensitivity experiments with different configurations where different variations of the methodology were tested. Below we discuss sensitivity to the choice of (1) prior covariance matrices, (2) observational targets, and (3) the number of



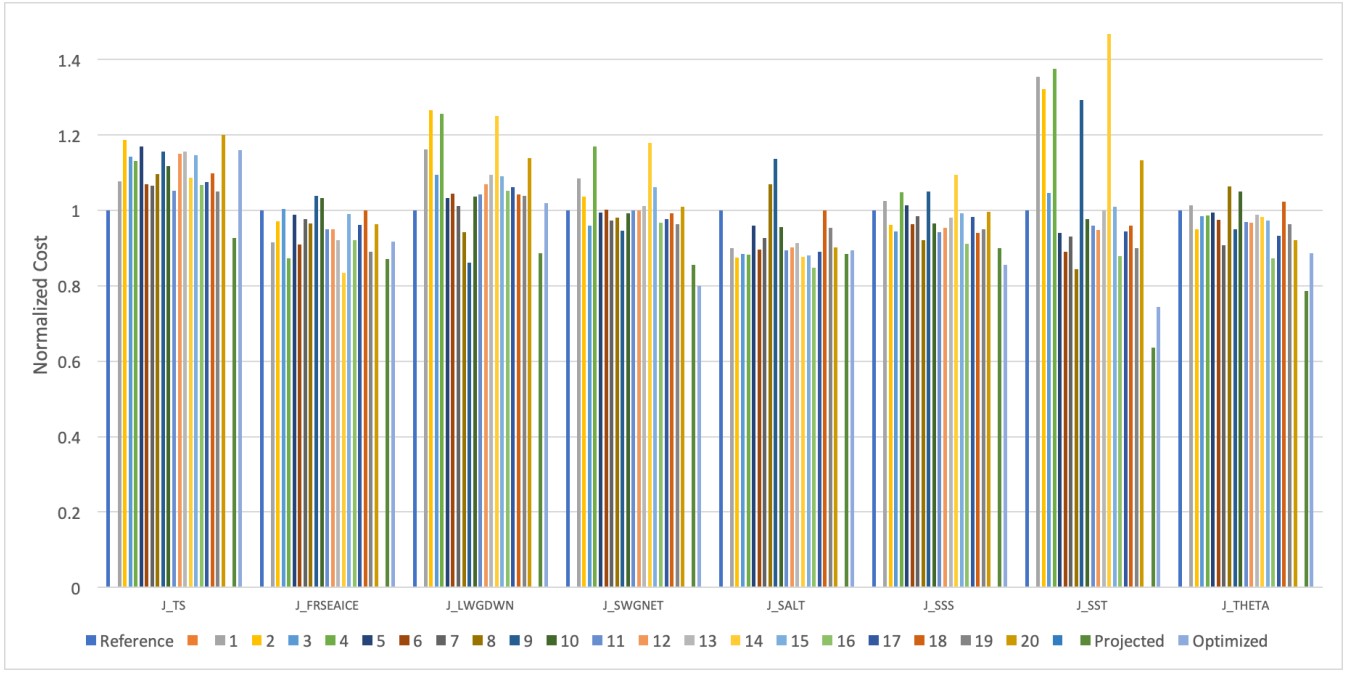

**Figure 2.** The cost of the reference experiment and the 20 perturbed experiments for each of the observational targets. Projected and optimized cost are also indicated.

control parameters. Many other sensitivity experiments can be designed. The goal of this section is not to cover all the options but rather to provide an additional dimensional depth of the Green's functions methodology. In the end, the future choice of the Green's function 'flavor' should reflect the goals of the optimization exercise.

## 5.1    Sensitivity to choice of prior covariance matrices

In order to evaluate the sensitivity to the choice of the covariance matrix, we used a simpler configuration, perturbing only three
parameters. The perturbed parameters were AUTscale, LTS_LOW, and two sea ice parameters (ALBICEV and ALBICEI) perturbed together. Using the Green's functions methodology, one can perform experiments in which more than one parameter is perturbed. The output from the methodology for this choice is a set of scaling factors for the parameters. When more than one parameter is perturbed in one experiment, the same scaling factor is applied to all the experiment parameters. In this configuration, unlike in our best practice configuration, the observational targets were multi-year means, and the cost was not
computed separately for each season. The observational targets are also similar to the best practice configuration except that the $500mb$ height was used in place of the surface temperature over land, and the net radiation replaced the two radiation components.





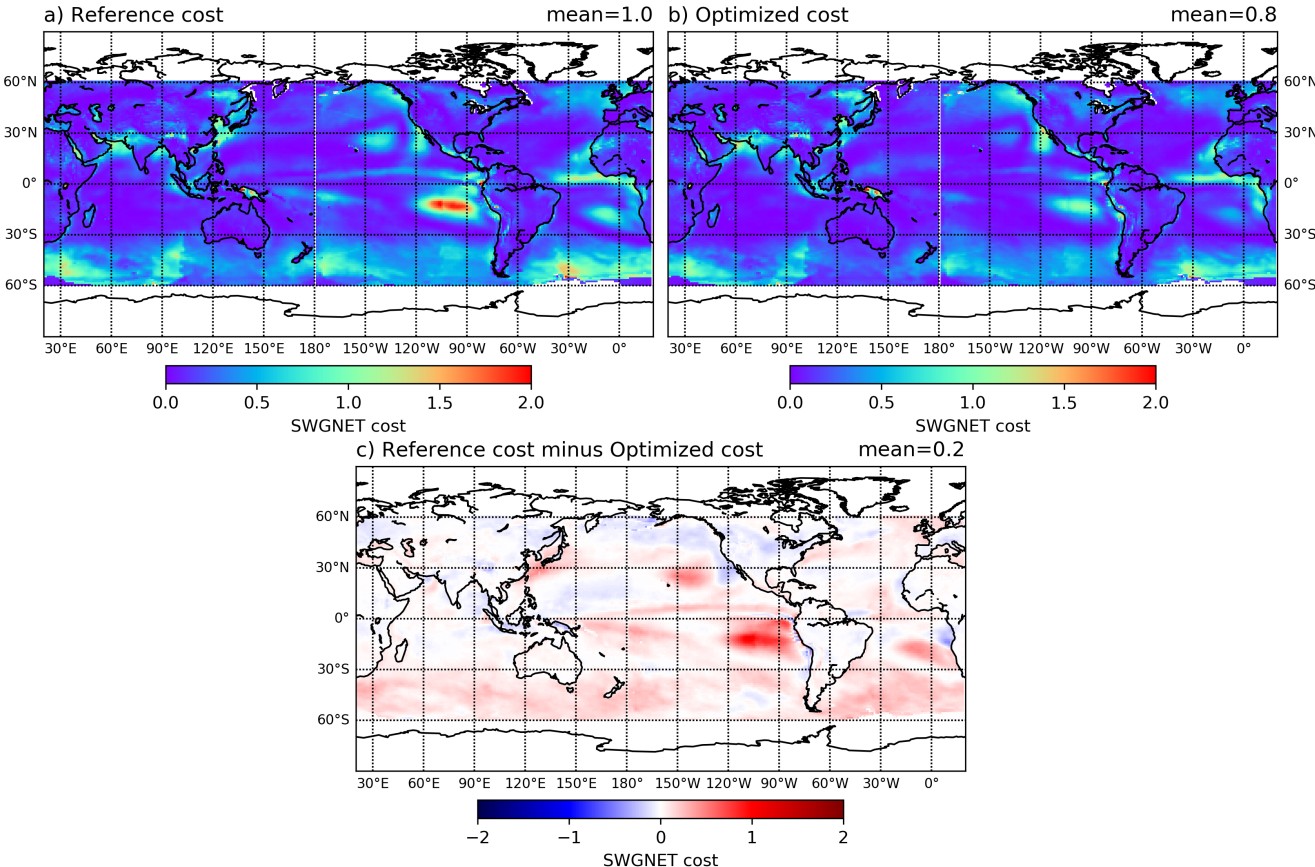

**Figure 3.** The net shortwave radiation cost for the reference (a), the optimized experiments (b), and their difference (c).

We tested four different options of the error covariance matrix:

$$R_v = \sum_{i,j,s} w_{i,j} \cdot \left( \hat{y}^o_{v,i,j,s} - \overline{\hat{y}^o_{v,s}} \right)^2 \ \ where \ \ \overline{\hat{y}^o_{v,s}} = \sum_{i,j} w_{i,j} \cdot \hat{y}^o_{v,i,j,s} \tag{11a}$$

$$R_{v,i,j,s} = \frac{1}{N} \sum_{t=1}^{N} \left( \tilde{y}^o_{v,i,j,s,t} - \overline{\tilde{y}^o_{v,i,j,s}} \right)^2 \ \ where \ \ \overline{\tilde{y}^o_{v,i,j,s}} = \frac{1}{N} \sum_{t=1}^{N} \tilde{y}^o_{v,i,j,s,t} \tag{11b}$$

$$R_v = \left( \max \left( \hat{y}^o_{v,i,j,s} - \hat{x}^r_{v,i,j,s} \right) \right)^2 \tag{11c}$$

$$R_v = \sum_{i,j,s} w_{i,j} \cdot \left( \hat{y}^o_{v,i,j,s} - \hat{x}^r_{v,i,j,s} \right)^2 \ \ (as \ in \ Eq. \ 10) \tag{11d}$$

where, $v, i, j, s, t$ are the variable, longitude, latitude, season, time indexes. The tilde ( ˜ ) represents seasonal means and $N$ is the number of years. The first option (a) uses the area-weighted variance of the observations instead of the mean-squared

misfits used in Eq. 10. The second option (b) is the spatially and seasonally variable, time variance of the observations, the third option (c) is the square of the maximal absolute difference between observations and the reference experiment, and the



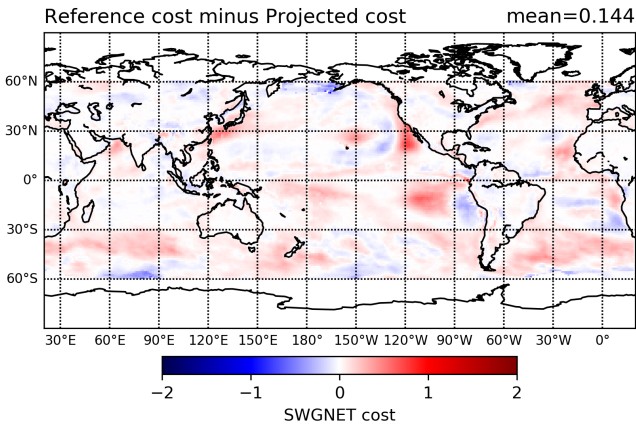

**Figure 4.** Net shortwave radiation cost difference between the projected value and the reference experiment

fourth option (d) is an area weighted variance of the difference between the observations and the reference experiment. All four options for the covariance matrix are calculated separately for each variable. We calculated the optimal parameters for each of the four definitions assuming $\mathbf{Q} = \infty$, that is, no prior information about the control parameters, and again assuming

that $\mathbf{Q} = \mathbf{I}$, the identity matrix, meaning that we have the same amount of confidence about the prior values of each control parameter.

Table 4 shows the eight different combinations of the parameters based on the four options for the observations covariance and two different parameters governing the covariance matrices. In general, all configurations generate realistic values in terms of the order of magnitude, but parameters differ in detail. The $\mathbf{Q} = \mathbf{I}$ case tends to keep parameters closer to their initial value

(relative to $\mathbf{Q} = \infty$) as expected.

Table 5 shows the cost associated with each of the eight experiments. Looking at the $\mathbf{Q} = \infty$ cases, defining the cost based on Eq. 11a reduce the cost by 7.7% but the relative importance of each variable varied substantially. For example, $500mb$ height cost was very small for the reference (0.024) and the fact that the optimized experiment increase the cost by 33.7% did not affect the total cost. Also, the sea ice contribution to the cost was much larger than the temperature and salinity at 300m

below the sea surface. Option 11b was found to provide only marginal cost reduction. Option 11c demonstrated the largest cost reduction (the smallest ratio in the J_total column). However, most of the projected cost reduction was restricted to the $500mb$ height target, while the cost of most of the other variables was not reduced, particularly in the case of the ocean targets. This is in contrast with option 11d, for which the cost reduction was spread more uniformly across the observational targets. All of these considerations led to our choice of option 11d as the best practice. The $\mathbf{Q} = \mathbf{I}$ was not chosen because the range of the

parameters is in itself often quite uncertain and because optimized parameter values were found to remain reasonable without the need for adding this additional constraint to the cost function. This choice could certainly be worth revisiting in a future study.



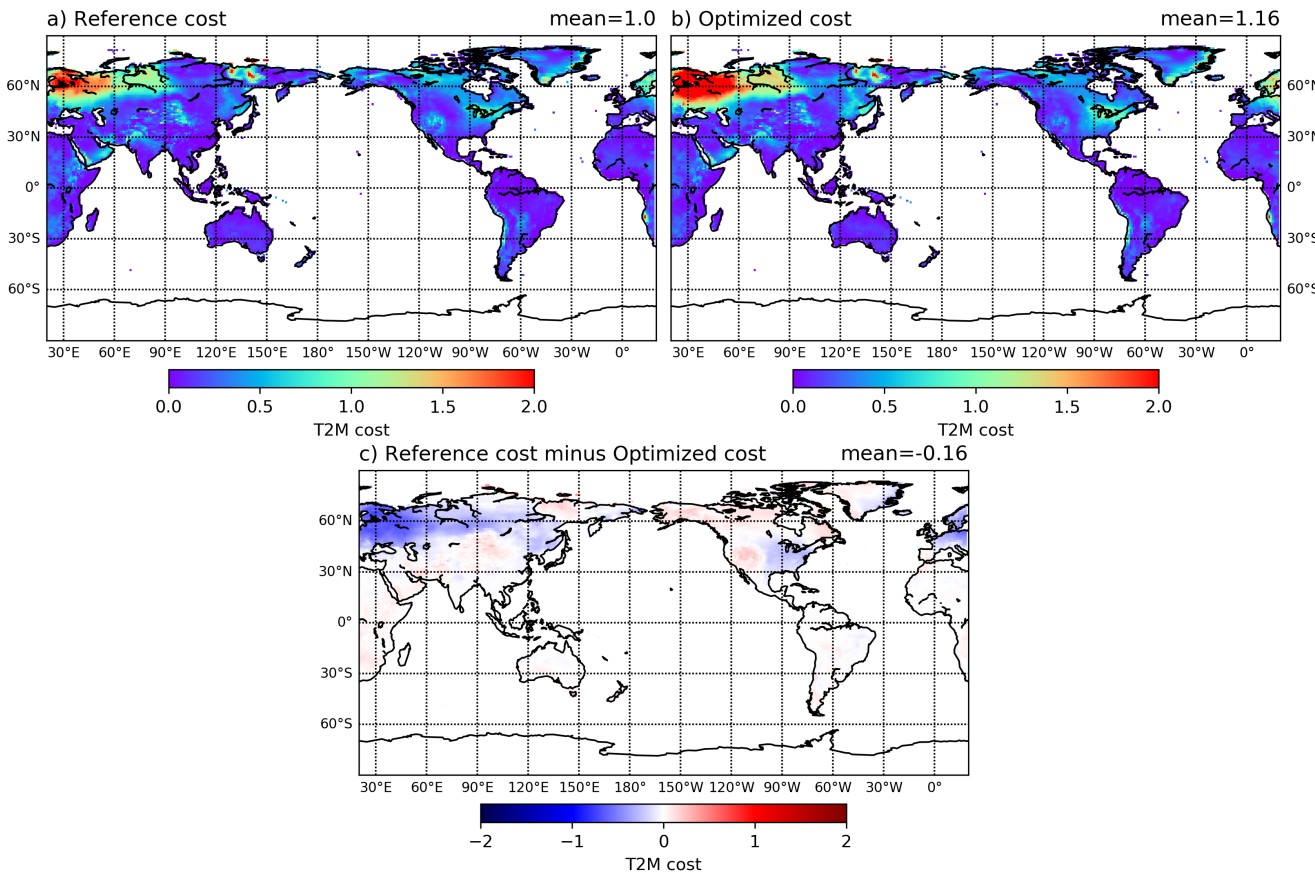

**Figure 5.** The 2 meter temperature cost for the reference (a) and the optimized experiments (b), and their difference (c).

After running a new model simulation with the optimized parameters based on Eq. 11d and $\mathbf{Q} = \infty$ case, the total cost reduction was found to be 6%, 2.9% less than the projected value, but still not small (Table 5). The $500mb$ height contributes
to most of the reduction, and the seaice cost was increased by about 25%. A possible explanation for the $500mb$ target being the main contributor to the cost reduction is that there were a small number of grid points in which the error was reduced substantially, particularly at high latitudes (not shown). This result suggests an experiment in which a spatially-dependant weight is used to reduce the magnitude of the outliers. The current cost implementation has the dependency on the model squared-error. This dependency favors the correction of a small number of points with large errors relative to a large number
of points with small errors. Choosing absolute error instead of squared-error would have changed the tendency to favor the correction of a small number grid points with large errors. Nevertheless, the simpler experiment configuration was a successful choice to explore the sensitivity to the covariance matrix designs.





**Table 4.** Optimized parameters of the 3 perturbed parameter experiments based on two different definitions of the parameters' covariance matrix **Q** and four definitions of the error covariance matrix (Eq. 11a-11d).

| Parameter → Experiment ↓ | | LTS_LOW | ALBICEV | ALBICEV | AUTscale |
|---|---|---|---|---|---|
| **Q** = ∞ | 11a | 18.68 | 0.846 | 0.420 | 0.241 |
| | 11b | 18.31 | 0.774 | 0.364 | 0.237 |
| | 11c | 17.79 | 0.871 | 0.440 | 0.116 |
| | 11d | 18.48 | 0.855 | 0.428 | 0.163 |
| **Q** = **I** | 11a | 18.51 | 0.734 | 0.333 | 0.201 |
| | 11b | 18.33 | 0.770 | 0.361 | 0.226 |
| | 11c | 18.34 | 0.749 | 0.345 | 0.138 |
| | 11d | 18.48 | 0.754 | 0.348 | 0.185 |
| ref. | | 18.5 | 0.73 | 0.33 | 0.2 |



**Table 5.** Cost of the 3 perturbed parameter experiments based on two different definitions of the parameter covariance matrix **Q** and four definitions of the error covariance matrix (Eq. 11a-11d).

| Parameter → / Experiment ↓ | | | J_total | J_SEAICE | J_H | J_RADSRF | J_SALT | J_SSS | J_SST | J_THETA |
|---|---|---|---|---|---|---|---|---|---|---|
| **Q = ∞** | 11a | reference | 0.885 | 0.294 | 0.024 | 0.106 | 0.038 | 0.370 | 0.017 | 0.035 |
| | | proj. / ref. | 0.923 | 0.909 | 1.337 | 0.969 | 0.896 | 0.886 | 1.076 | 0.973 |
| | 11b | reference | 445.7 | 72.1 | 14.9 | 30.4 | 93.1 | 74.9 | 42.3 | 117.8 |
| | | proj. / ref. | 0.987 | 0.964 | 1.058 | 1.015 | 0.976 | 0.970 | 0.995 | 1.003 |
| | 11c | reference | 4.283 | 0.106 | 4.049 | 0.052 | 0.007 | 0.004 | 0.039 | 0.027 |
| | | proj. / ref. | 0.445 | 1.356 | 0.398 | 1.174 | 1.255 | 0.974 | 1.160 | 1.168 |
| | 11d | reference | 7 | 1 | 1 | 1 | 1 | 1 | 1 | 1 |
| | | proj. / ref. | 0.921 | 1.099 | 0.689 | 0.966 | 0.997 | 0.872 | 0.844 | 0.979 |
| | | **opt. / ref.** | **0.940** | **1.253** | **0.523** | **1.045** | **1.010** | **0.864** | **0.950** | **0.937** |
| **Q = I** | 11a | reference | 0.885 | 0.294 | 0.024 | 0.106 | 0.038 | 0.370 | 0.0166 | 0.0354 |
| | | proj. / ref. | 0.961 | 0.924 | 1.199 | 0.991 | 0.935 | 0.964 | 1.048 | 0.986 |
| | 11b | reference | 445.7 | 72.1 | 14.9 | 30.4 | 93.1 | 74.9 | 42.3 | 117.1 |
| | | proj. / ref. | 0.987 | 0.964 | 1.056 | 1.015 | 0.976 | 0.970 | 0.995 | 1.003 |
| | 11c | reference | 4.283 | 0.106 | 4.049 | 0.052 | 0.007 | 0.004 | 0.039 | 0.027 |
| | | proj. / ref. | 0.484 | 1.289 | 0.444 | 1.087 | 1.215 | 0.953 | 1.080 | 1.084 |
| | 11d | reference | 7 | 1 | 1 | 1 | 1 | 1 | 1 | 1 |
| | | proj. / ref. | 0.926 | 1.062 | 0.703 | 0.972 | 1.007 | 0.898 | 0.859 | 0.982 |

## 5.2 Sensitivity to choice of number of parameters

Here, we evaluate the sensitivity of the cost to the number of optimized parameters. We calculated the projected cost reduction for four cases which used a subset of the first 5, 10, 15 and all 20 parameters from Table 2. We focus here on an optimization based on the multi-year means. An increased number of parameters is expected to translate into a reduced cost as we fit the model to more parameters. In the following results we decided to remove the 500mb height from the cost function in order to spread the cost across more variables. Generally, there is nothing wrong with having most of the cost reduced in one of observational targets, but we decided here to seek a more balanced cost reduction. An additional change here and in the following sections is the separation of the surface radiation targets into net shortwave and downward longwave radiation, as both have SRB observational counterparts.

We tested the projected cost reduction as a function of the optimized parameters' number (figure 6). We found that the projected cost was reduced from 6% to 25% when going from 5 optimized parameters to 20. The cost reduction was gradual





showing and additional reduction with the increase of the optimized parameters. The largest cost reduction was found between
5 to 10 optimized parameters.

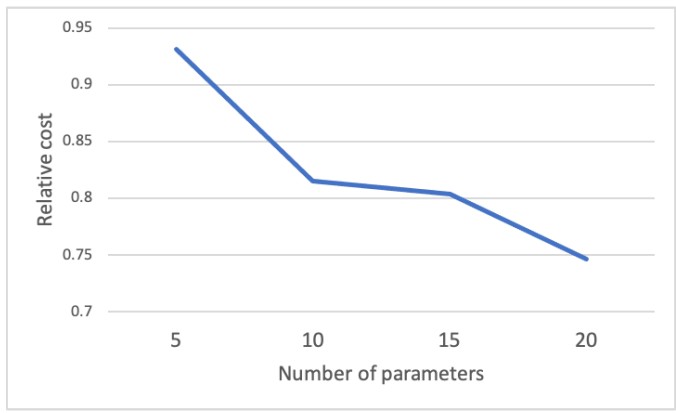

**Figure 6.** Cost relative to the reference experiment as a function of the number of optimized parameters based on annual means.

### 5.3   Annual-Mean versus Seasonal-Mean Data Constraints

Here, we used nine sensitivity experiments, with nine optimized parameters, to look at the sensitivity of the optimized parameters to seasonality. Two sets of parameters were calculated - based on multi-year means (annual, Table 6) and based on multi-year seasonal means (seasonal, Table 6). The calculation of the optimized parameters comes after the simulations were
performed, so the input simulations are the same for both options. The only difference between the annual and seasonal suite is the configuration of the cost function provided to the Greens' functions methodology.

The most noteworthy difference between the annual and seasonal results was found in the Charnock parameter that controls surface winds. Although the difference between parameters is smaller than 10%, the projected cost differences are large, with the annually-based optimization reducing the cost by about 15% and the seasonally-based optimization reducing the cost by
about 9%. This result is expected, as seasonally-based data have more variability and we use the same number of parameters to optimize model results to larger number of observations.

Optimizing for seasonal observational targets is expected to be essential for improving intra-annual variability. Therefore, despite the less efficient cost reduction for the seasonal targets we decided to continue to focus on experiments with seasonal observational targets. For these two sets of experiments, we did not choose to re-run the model with optimized parameters, but
instead decided to increase the number of optimized parameters by performing additional simulations.

### 5.4   Length of Optimization Period

This sensitivity test was done with the 20-parameter set of simulations. We performed two Green's function calculations: one that optimizes the parameters based on the last five years of the runs (2005-2010) and another that uses the whole ten years (2000-2010) as in section 4. The motivation here was to avoid using years during the model drift after the initialization. The





**Table 6.** Optimized parameters of the 9 perturbed parameter experiments configured with $Q = \infty$ and option 11d.

| Parameter → Experiment ↓ | AUTscale | LTS_LOW | CICE_AH_MAX | ALBICEV | ALBICEI | TURN_RHCRIT | MAX_RHCRIT | CQfactor | Charnok1 |
|---|---|---|---|---|---|---|---|---|---|
| Reference | 0.2 | 19 | 0.3 | 0.73 | 0.33 | 884 | 1.0 | 1.0 | 2.92E-3 |
| Perturbed | 0.3 | 20 | 0.2 | 0.82 | 0.40 | 750 | 0.8 | 1.5 | 2.19E-3 |
| annual | 0.194 | 19.53 | 0.354 | 0.834 | 1.14 | 850 | 1.14 | 1.25 | 3.54E-3 |
| seasonal | 0.203 | 19.38 | 0.334 | 0.797 | 1.12 | 865 | 1.12 | 1.23 | 3.29E-3 |

**Table 7.** Proj. / ref. cost of the 9 perturbed parameter experiment configured with $Q = \infty$ and option 11d.

| Target → Experiment ↓ | J_total | J_SEAICE | J_H | J_RADSRF | J_SALT | J_SSS | J_SST | J_THETA |
|---|---|---|---|---|---|---|---|---|
| annual | 0.854 | 0.822 | 0.917 | 0.873 | 0.938 | 0.978 | 0.634 | 0.815 |
| seasonal | 0.912 | 0.887 | 0.933 | 0.899 | 0.935 | 0.973 | 0.922 | 0.835 |

2005-2010 optimization projected a 12.4% cost reduction compared to a 15.6% projected cost reduction for the 2000-2010 experiment. The optimized experiment with the optimized parameters had a cost reduction of 7.1% relative to 9% of the 2000-2010 experiment. One of the plausible reasons for less cost reduction is that 2005-2010 had less than two El ñino cycles, and therefore it was too short for the calculation of the model climatology. The 10-year experiment had three full El ñino cycles, and its climatology represents the model climatology more adequately. The 10-year optimization, therefore, is found to enable

better parameter optimization.

## 6  Summary and Concluding Remarks

This study demonstrates the applicability of the Green's functions approach, introduced in Menemenlis et al. (2005) for a forced ocean simulation, to the adjustment of uncertain parameters for a coupled ocean-atmosphere simulation. In this proof-of-concept study, we computed the response of the coupled ocean-atmosphere simulation to the perturbation of the 20 parameters

listed in Table 1. These 20 perturbation experiments were subsequently used to optimize the values of the 20 parameters. The observational targets were long term seasonal-means of the eight fields listed in Table 2. The optimization increased the explained variance by up to 25% (for SST in Table 3).

The Green's functions approach assumes that the response of the model to small perturbations is approximately linear. This assumption does not rigorously hold for atmospheric, oceanic or coupled-ocean-atmosphere models because of non-linear

weather and climate phenomena such as synoptic storms and El Niño-Southern Oscillation (ENSO) variability. Therefore, a key consideration is that the perturbation experiments be sufficiently long to average out non-linear weather and lower frequency phenomena. This study shows that 10-year long simulations can be sufficient to enable successful application of the Green's





functions approach. However, 10 years may still be too short to fully average out some important low frequency variability. This causes differences between the projected linear combination and the optimized simulation, shown as the differences in
the explained variance reported on Table 3. This also causes some degradation relative to the baseline simulation for two of the observational targets, the near-surface temperature (TS) and downward longwave radiation (LW).

The cost reduction was spread across most of the observational targets, though our configuration of observational targets may have been responsible for the cost increase in two amongst them — the targets associated with land areas that are under-represented in our cost function. In the end, The choice of the observational targets should reflect the objective of the study and,
probably, there is no single set of observational targets that is adequate for all applications *a priori*. Our proof-of-concept study and the experiments that show the impact of different choices in the details readily reflect the trade-offs among data constraints that can be important considerations.

Increasing the number of optimized parameters reduced the overall cost which seems to be quasi-linear with the number of parameters, but further investigation into this is required. Increasing the temporal resolution of the observational targets
from annual to seasonal increased the cost further – a result of the increasing number of the observational targets. Adding seasonality in parameter adjustments is viewed as one of the next logical steps in this regard. We also tested four different methods to calculate the error covariance matrix, and found that using the error of the reference experiment produced the most balanced results in terms of accounting for all observational targets.

This study shows that the Green's functions methodology can benefit the climate modeling community by providing a more
structured yet practical method to tune climate models. The Green's functions methodology has several advantages that are worth emphasizing here: (a) It is simple to implement and it does not require internal amendments to the model code. (b) It is cheap for small number of parameters – one extra sensitivity experiment per parameter. (c) One can generate virtual predictions and calculate projected cost for infinite number of cost configuration without running new experiments. (d) It offers a systematic way to optimize parameters accounting for a possible dependent response of the model to a change of different
parameters. (e) The optimization process can be extended with more sensitivity experiments and it can be also revisited with different set of parameters. (f) The sensitivity experiments can be done in parallel suggesting potential scalability.

Identifying the most important uncertain parameters for applying Green's functions methodology still requires close famil-iarity with models and there is no replacement for the experience of the modelers when using this methodology. Therefore it is viewed as a valuable framework to further leverage modelers' expertise to fine-tune model parameters, standardize practices
across the various climate modelling centers, and improve climate model products beyond the present state of art.

*Code and data availability.* Data leading to this publication was made available as part of a Zenodo repository at:

– https://doi.org/10.5281/zenodo.5507631

This repository includes models code and configuration files, post-processed data from model simulations and observational targets, and Green's functions-related scripts, including those generating the figures and tables.



*Author contributions.* ES ran the simulation and performed the experiments. AM, DB, AT, DM and GF contributed to the design of the experiments and writing the manuscript.

*Competing interests.* The authors of the manuscript declare that no competing interests are present

*Acknowledgements.* – The SRB data were obtained from the NASA Langley Research Center Atmospheric Sciences Data Center NASA/GEWEX SRB Project.



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
