# Peer review of "Earth System Model Parameter Adjustment Using a Green's Functions Approach"

_Geoscientific Model Development, 2021_

## Author Comment (AC1)

We thank the reviewers for their positive feedback. In what follows we provide a breakdown of our response to the reviewer comments. The text in black is the reviewer's comment, the text in green is our response to the reviewers, and the text in red represents text incorporated into the revised manuscript.

**Reviewer 1**

This papers proposes an objective method to tune ESMs in a global approach. The authors proposes a mathematical equivalent to the way modellers tune their models so as to rationalize and automatize the tuning process, based on a Green's functions approach. This is a very important topic for ESM developers that come to a limitation in the model developments due to the long process of human-made iterative tuning. This study is perfectly timely. The authors provides an example of application and discuss some of the assumptions made. However, some assumptions are not fully explained or would deserve more comments. I therefore recommend the manuscript to be accepted after minor revisions.

**Main comments**

Section 3: Explaining the methodology using a mathematical formalism is a very good idea, but in the present form, I have not the impression that it is sufficiently clear. Maybe I am not a good mathematician but I think that this manuscript should be well understood by modellers and improving the explanation would leverage the impact of the study. Moreover, the equations provided in the manuscript are already presented in Menemenlis et al. (2005), therefore, they do not provide anew outlook. I would suggest to better explain these equations here.

We have modified the explanation of the equations and added more information to make the section clearer as detailed below.

It is not clear to me how equation (4) is established?

We added:

"To fit the ESM to the available observations, we aim to minimize an objective cost function

**J = ... (4)**

which penalizes a weighted least-squares distance between optimized and prior model parameters ( $\eta^{T} \mathbf{Q}^{-1} \eta$ ) and a weighted least-squares distance between simulation and observations ( $\epsilon^{T} \mathbf{R}^{-1} \epsilon$ ), where T is the transpose operator and -1 indicates

matrix inversion. It can be shown that the minimization of Equation (4) will produce the maximum likelihood solution for Gaussian inverse problems (Kalnay, 2002)."

Similarly, it is not clear to me what is the difference between G(AE) and GAE? Maybe the notations could be simplified.

We added the following explanation:

"... and kernel matrix **G** can be thought of as a first order multivariate Taylor expansion of *G* around  $\eta$ =0. The assumption here is that for small enough perturbation, the model response is linear."

Additionally, the physical meaning of Q and R should be detailed as well as the methodology to set them.

We added the following paragraph in Section 5.1:

"In general, the error covariance matrices **Q** and **R** are not known and difficult to estimate. For the Green's function approach, however, the number of observations is generally much larger than the number of parameters being estimated, which simplifies the selection of **Q** and **R**. First, the small number of control parameters limits the solution's degrees of freedom; therefore the choice of **Q** and **R**, if they are reasonable, is not expected to change the solution of  $\eta^a$  much – but it will impact posterior uncertainty **P**. Second, the data kernel matrix **G** is small enough to be defined explicitly; therefore many interesting properties of the solution can be derived and evaluated. Third, the solution of (8) and (9), once the kernel matrix **G** has been derived, is trivial; therefore it is possible to test the impact of particular choices of **Q** and **R**, as is done next."

For instance in section 4.3, it is said that "R is chosen diagonal": apart from being mathematically convenient, what are the physical hypothesis behind such a choice? What makes this choice sensible? Isn't it a strong hypothesis? What happens if reality is far from diagonal? These points should be discussed with a physical point of view in the manuscript to better illustrate the advantages and flaws of the method used.

We added information about the motivation of choosing diagonal **R** in section 4.3:

Because the estimation problem is highly overdetermined --- the number of parameters to adjust is O(10) while the number of observations is O(10^6) --- we expect that the assumption that **R** is diagonal will have minimal impact on the optimized estimate  $\eta^a$ . In general, the addition of non-diagonal elements of **R** would indicate linear dependence among observational errors, reducing the effective degrees of freedom. Since, in our case, the number of observations is far larger than the number of optimized parameters,

we would still retain a very large number of independent observations as compared to optimized parameters.

Similarly, section 5.1 is a bit hard to read.

The start of this section was modified to clarify the motivation for using a simplified configuration. In addition to the paragraph explaining the physical meaning of **Q** and **R** (see above), we added:

The sensitivity to the choice of covariance matrices  $\mathbf{Q}$  and  $\mathbf{R}$  in Equation (4) is evaluated using a pared-down configuration, perturbing only three parameters. The pared-down configuration is used to simplify the discussion and better illustrate the response of the optimized solution to different covariance matrices.

The physical idea behind the choice of functions is not clear enough. What is the physical meaning of Q choices proposed? This section need to be detailed to better catch the physical hypothesis that are made depending on each choice of covariance matrix.

The choices for **R** and **Q** represent plausible choices for **R** and extreme choices for **Q**. We added:

The four estimates of **R** aim to represent a range of plausible estimates for the actual error variance and allow us to explore the effect on the resulting parameters and cost.

We calculated optimized parameters  $\eta^a$  for each of the four definitions of **R** assuming that **Q**= $\infty$ , that is, no prior information about the control parameters, and again assuming that **Q**=**I**, the identity matrix, meaning that we have the same amount of confidence about the prior values of each control parameter. These two estimates of **Q** represent two extreme situations.

There is also a crucial point that should be at least discussed in the paper: in such complex ESM there is a long spin-up process of several decades that has to be taken in to account. If only taking the first 10 years of ESM simulations for tuning, what is the result of such a sort term tuning on the long term equilibrium? How do the authors proposes to handle this problem?

Text has been added to the manuscript to clarify the rationale for the choice of experiment length:

"The length of the simulations needed for the Green's Function optimization is related to the application of the model being optimized. For climate projection applications, where the long-term equilibrium of the model is paramount, it seems clear that longer simulations would be warranted in order to allow for model spin up time. The primary application of the coupled model presented here is seasonal to decadal prediction, and so the optimization is done to include (ideally to minimize) the initial model drift."

For such an application the authors feel that 10 year simulations are adequate.

Line 230: the fact that the surface temperature is less well reproduced after the tuning process is very interesting. Even if it reflects a flaw in the present choice of observational targets, the discussion on this should be enriched. What type of other observational targets could be proposed to avoid this?

We suggest balancing the observational target by including more observations specific to the land, e.g., snow fraction and soil moisture. Text clarifying this was added to the paragraph:

"Including more land-related observational targets such as snow fraction and soil moisture could improve the cost of surface temperature."

Section 5.2: by increasing the number of parameters, there is a risk to over-fit the model to the set of observation chosen as discussed in Dommenget and Rezny, (2018). This risk should be discussed in the last section. Additionally, by increasing the number of parameters the final fit is necessary improved as more degree of freedom have been used. Do the authors could use a methods similar to the LASSO technique (Tibshirani, 1996, used in linear regression) to penalize the results according to the number of degree of freedom? This would help selecting only the more useful parameters and avoid over-fitting.

There is no risk of overfitting since we optimize about 20 parameters, whereas the observational target includes hundreds of thousands of "observations". We have about eight observational targets, each having 10,000 grid cells times four seasons (~320,000 grid cells). Therefore, the Green's function methodology will not overfit the model parameters. We have not explored methods such as LASSO because of the small number of control parameters relative to the number of degrees of freedom.

To clarify this point, we added additional text at the end of section 4.4:

"Note that there is no risk of overfitting due to the large number of observational targets (eight observational targets, two-dimensional fields, four seasons≅300,000 targets). Methods to penalize a large number of parameters are therefore not needed in this case."

Ln 304-305: "The largest cost reduction was found between 5 to 10 optimized parameters." And what about the ratio projected cost/optimized cost? Maybe this information could be added on figure 6. Is this ratio stable depending on the number of parameters?

We do not have the optimized simulation for this set of experiments. Our motivation here was to demonstrate that much information can be gained just by calculating the projected cost. In future optimization exercises, this can help understand the effect of adding more optimized parameters. The considerable cost reduction between 15-20 motivates an increase in the number of parameters in the future.

Section 5.4: what would be the cost if the first 5 years were used? Would it be different from the last five years? If yes, this would probably says that the spin-up is important. But then, what about taking the last 10 years over a 20 years simulation. The initial choice of 10 years appears as a bit short against the spin-up timescale of coupled models. The spin-up problem would be discussed more thoroughly. Maybe it would be nice to show the model initial drift.

We have recalculated the cost for the first 5 years of the simulations and found it to be different from the last five years (as there was a difference between the last five year cost and the full 10-year cost as discussed in section 5.4). The following text was added to section 5.4:

"The 2000-2005 optimization projected a 15.3% cost reduction, the 2005-2010 optimization projected a 12.4% cost reduction, and the 2000-2010 optimization projected a 15.6% cost reduction."

We agree with the reviewer that the spin-up is important. The choice of 10-year simulations was in part motivated by computational cost, but was largely motivated by the application of the model that is being tuned. If the application is climate projection, then we agree that 10-year simulations, which would be influenced by the spin-up, would be quite short. However, for applications such as seasonal or decadal prediction, it would be important to include the spin-up phase in the optimization - each forecast will undergo a spin-up phase as it drifts from the observed initial state.

The following text was added to the main manuscript to address this reviewer's earlier comment about the spin up and is relevant in response to this comment as well:

"The length of the simulations needed for the Green's Function optimization is related to the application of the model being optimized. For climate projection applications, where the long-term equilibrium of the model is paramount, it seems clear that longer simulations would be warranted in order to allow for model spin up time. The primary

**application of the coupled model presented here is seasonal to decadal prediction, and so the optimization is done to include (ideally to minimize) the initial model drift."**

I may have misunderstood the methodology, but reading again the introduction section, I am not convinced that the proposed methodology addresses all the points raised in the introduction.

We agree. In the revised manuscript we are more careful with explaining what the approach can and cannot do.

- In 49-52 :"An experiment with both perturbed would not reveal which had the greater impact on the solution, and experiments with only one at a time perturbed would not behave as an experiment with both perturbed. The sequential methodology, therefore, for this example, may result in a sub-optimal combination of these parameters."I do not understand how the Green's function approach do something different to "one at a time" perturbed experiment. Maybe it is just a question of formulation here?

The text of the introduction was modified to explain the difference between the "one at a time" approach and the Green's functions approach:

"As is explained in section 3, the Green's functions methodology includes a single suite of simulations where each parameter is perturbed one at a time, as in a sequential method, but performs the optimization by adjusting all of parameters simultaneously while accounting for the potential interdependence of their effect on the simulation. The optimization step can be repeated using different observational targets with no additional computational expense. The impact of the parameters is assumed linear, but given the limits of that assumption the result is the optimal combination of all the parameters at once. This approach accounts for the combined impact of all the parameters while also establishing the individual impact of any one parameter on the solution and with respect to each observational target."

- In 58 :"The second drawback is the large number of needed simulations, since a new set of experiments is required for each set of observational targets". Using the standard by hand method, developers usually have a look at a panel of diagnostics and do not need to redo the experiments for a new diagnostic, I do not see any difference on this point. This point is repeated on line 74, I would not emphasize this much.

The text in the paragraph of the introduction discussing the Green's functions approach addresses this comment of the reviewer:

"An iterative sequential tuning method, on the other hand, would require several suites of sensitivity experiments, each dictated by the results of the suite before." - In 71-71: "The interdependence is accounted for in the constraints on the error covariance for each parameter." This points of interdependence is not well developed in the manuscript. I would promote to better illustrate this point.

This statement was removed from the manuscript, as the paragraph now includes the explanations given in response to the reviewer's previous comments.

**Miscellaneous**

Section 2: please indicate the ocean model time-step and better separate the ocean and atmosphere models description in two distinct paragraphs.

**Done**

Table 4 and 5: these tables are large and difficult to interpret as is, the exact values not behind of strong interest. Please consider replacing them by plots.

We tried the plot but found the table clearer. We believe that the exact numbers provide additional value in this case.

Section 5.2: Concerning LW and SW radiation, is the change from downward to net? Why changing this point? Could the authors discuss a bit more this choice?

Earlier experiments used the net radiation as the observational target, but we split it into downward longwave and net shortwave due to their independence. For the longwave radiation, it is only the downward component that acts as forcing to the ocean.

The text in section 5.2 now reads:

"An additional change here and in the following sections is the separation of the net surface radiation target into net shortwave and downward longwave radiation, as they are independent and both have SRB observational counterparts. The choice of downward longwave rather than the net was made due to the direct dependence of upward longwave radiation on the SST (which is already a constraint)."

References :

Dommenget, D. and Rezny, M.: A Caveat Note on Tuning in the Development of Coupled Climate Models, J. Adv. Model. Earth Syst., 10, 78–97, https://doi.org/10.1002/2017MS000947, 2018.

Tibshirani, R.: Regression Shrinkage and Selection Via the Lasso, J. R. Stat. Soc. Ser. B Methodol., 58, 267–288, https://doi.org/10.1111/j.2517-6161.1996.tb02080.x, 1996. Citation: https://doi.org/10.5194/gmd-2021-251-RC1

**Reviewer 2**

In this work the authors used Green's function approach to optimize parameters in an earth system model. This optimization was achieved by minimizing a weighted least-squares distance. The numerical results demonstrated that the Green's functions estiamtion approach can be a good fine-tuning step in the model development process to adjust uncertain parameters of the model considered in this work. Also, sensitvity analysis results were presented to further investigate the effectiveness of the Greens's function approach. In general, as an appliction work, the manuscript is well organized and technical details are clear.

I have a few comments on the approach itself although this work is focused on application.

1. Eq(5) is the combination of Eq (1) and Eq (3), and G seems like a composite of  $M_i$  and H. How do we understand it as an "convolution of observation operator H and  $M_i$ "?

Convolution was used loosely to describe "a mathematical operation on two functions that produces a third function". The reviewer is correct that the terminology was misleading so we replaced "convolution" with "combination".

2. Eq(4) and Eq (9) are similar to some formulas in Kalman filter. Indeed, Eqs (2) and (3) are also the starting point of Kalman filter. What is the main difference between the Green's function method and Kalman filter? Or are they equivalent to each other to some extent or in some cases?

The reviewer is correct. We added the following sentence in the revised manuscript: "Mathematically, the Green's function approach is equivalent to any linear, Gaussian inverse problem, for example, the adjoint method." Because the Kalman filter carries information forward in time, it cannot be used to estimate model parameters that are applied for the complete period of integration.

3. The impact of choosing different Q was discussed using Q=\infty and Q=I. How about the influence of R? Here R is a diagonal matrix, which indicates that the errors for different observables are independent (which is a quite standard setup in Kalman filter). How about the errors are not independent? Any comments or intuitions? Citation: https://doi.org/10.5194/gmd-2021-251-RC2

This same question was raised by Reviewer 1. The following discussion was added to the manuscript in Section 4.3:

Because the estimation problem is highly overdetermined --- the number of parameters to adjust is O(10) while the number of observations is O(10A6) --- we expect that the assumption that **R** is diagonal will have minimal impact on the optimized estimate  $\eta^a$ . In

general, the addition of non-diagonal elements of **R** would indicate linear dependence among observational errors, reducing the effective degrees of freedom. Since, in our case, the number of observations is far larger than the number of optimized parameters, we would still retain a very large number of independent observations as compared to optimized parameters.